# Advances in the Antagonism of Epigallocatechin-3-gallate in the Treatment of Digestive Tract Tumors

**DOI:** 10.3390/molecules24091726

**Published:** 2019-05-03

**Authors:** Changwei Liu, Penghui Li, Zhihao Qu, Wei Xiong, Ailing Liu, Sheng Zhang

**Affiliations:** 1Key Laboratory of Tea Science of Ministry of Education, Hunan Agricultural University, Changsha 410128, China; liuchangwvv@163.com (C.L.); lpflstt111@163.com (P.L.); 18579546687@163.com (Z.Q.); 2National Research Center of Engineering Technology for Utilization of Functional Ingredients from Botanicals, Collaborative Innovation Centre of Utilisation of Functional Ingredients from Botanicals, Hunan Agricultural University, Changsha 410128, China; 3The Key Laboratory of Carcinogenesis and Cancer Invasion of the Chinese Ministry of Education, Cancer Research Institute, Central South University, Changsha 410078, China; xiongwei@csu.edu.cn; 4College of Bioscience and Biotechnology, Hunan Agricultural University, Changsha 410128, China

**Keywords:** epigallocatechin-3-gallate, digestive tract tumor, colorectal cancer, apoptosis, cellular signaling cascades, anti-cancer therapy

## Abstract

Due to changes in the dietary structure of individuals, the incidence of digestive tract tumors has increased significantly in recent years, causing a serious threat to the life and health of patients. This has in turn led to an increase in cancer prevention research. Many studies have shown that epigallocatechin-3-gallate (EGCG), an active ingredient in green tea, is in direct contact with the digestive tract upon ingestion, which allows it to elicit a significant antagonizing effect on digestive tract tumors. The main results of EGCG treatment include the prevention of tumor development in the digestive tract and the induction of cell cycle arrest and apoptosis. EGCG can be orally administered, is safe, and combats other resistances. The synergistic use of cancer drugs can promote the efficacy and reduce the anti-allergic properties of drugs, and is thus, favored in medical research. EGCG, however, currently possesses several shortcomings such as poor stability and low bioavailability, and its clinical application prospects need further development. In this paper, we have systematically summarized the research progress on the ability of EGCG to antagonize the activity and mechanism of action of digestive tract tumors, to achieve prevention, alleviation, delay, and even treat human gastrointestinal tract tumors via exogenous dietary EGCG supplementation or the development of new drugs containing EGCG.

## 1. Introduction

Gastrointestinal tumors represent sarcomas of the digestive tract and its accessory organs, such as oral cancer, esophageal cancer, gastric cancer, colorectal cancer, and liver cancer. Most tumors of the gastrointestinal tract are malignant epithelial tumors [1], and the main pathological type, adenocarcinoma or squamous cell carcinoma, can be destructive and can seriously threaten the life and health of a patient. Many factors such as chemical [2], physical [3], biological [4] and behavioral factors [5] trigger the development of gastrointestinal tumors. If *Helicobacter pylori* infection can result in gastric cancer, long-term drinking can easily result in gastric cancer and liver cancer. The complex causes of tumors have led to a significant increase in the incidence of digestive tract cancers. In the 2018 American Cancer Report released in early 2018 [6], there were 1,735,350 new cancer cases in the United States, with an average of 4700 cases reported per day. Among them, 140,250 patients had colorectal cancer, ranking this tumor in the top three. In the 2018 National Cancer Report released by the National Cancer Center in China [7], the estimated number of new cases of malignant tumors in 2014 was 3.804 million, of which gastric and colorectal cancer ranked second and third. The prevalence of colorectal cancer ranked second and third. Gastrointestinal tumors have therefore become one of the most aggressive tumors affecting the wellbeing of modern humans.

Epigallocatechin-3-gallate (EGCG) in green tea has attracted the attention of researchers because it can interrupt the production and development of various tumors such as breast cancer [8], lung cancer [9], liver cancer [10] and colorectal cancer [11]. EGCG can also reduce the recurrence rate of cancers which has attracted research interest [12]. The anti-cancer mechanism of EGCG involves angiogenesis inhibition, tumor cell death induction and tumor growth inhibition. Many recent studies have shown that EGCG can prevent the development of colorectal cancer by eliminating inflammatory factors [13] and can induce apoptosis of gastric cancer cells by regulating cellular metabolic pathways [14,15]. Its combination with curcumin has also been reported to inhibit gastric angiogenesis [16]. EGCG can enhance the anticancer activity of other anticancer drugs [17], reverse cell resistance against cancer drugs and reduce the likelihood of recurrence after tumor surgery. Altogether, a series of studies have shown that EGCG displays a good effect when used to prevent and antagonize digestive tract tumors [18,19]. EGCG can also be administered orally, resulting in direct contact with the digestive tract epithelial cells and the localization of most of its content in the gastrointestinal tract. EGCG possesses acceptable safety [20], and from the perspective of economic cost, it can be efficiently prepared from tea [21], which contributes to a low medical cost. These attributes have contributed to the potential of EGCG as a drug against gastrointestinal cancer. In this article, we intend to summarize the mechanism of antagonized tumor action by EGCG, the antagonizing effect of its derivatives on digestive tract tumors, and its synergistic anti-cancer effect with other medications. We also aim to provide a reference for the development of EGCG as a drug substance to prevent and antagonize digestive tract tumors.

## 2. EGCG

### 2.1. Physical and Chemical Properties

Catechin is the main secondary metabolite of *Camellia sinensis* (L.) O. Kuntze, accounting for an estimated 12% to 24% of the dry weight of tea, and EGCG is the primary content, accounting for approximately 50–80% of the total amount of catechins [22]. EGCG is a derivative of 2-phenyl benzo, which consists of three necessary rings (A, B and C), and a gallic acyl group containing a D ring. Many phenolic hydroxyl groups are distributed on its A, B and D rings, and importantly, three ortho-phenolic hydroxyl groups exist on the B and D rings, allowing the strong antioxidant capacity and free radical scavenging ability [23] (Figure 1). The number and location of the gallic acyl moiety and the hydroxy group on the catechin ring affect the pharmacological properties of EGCG [24].

### 2.2. Bioavailability of EGCG

Due to the diverse biological activities of EGCG, in vitro and in vivo experiments have shown that EGCG exerts its biological activity by regulating a variety of signaling pathways. However, complete absorption of EGCG does not occur in the gastrointestinal tract; the amount reaching target organs/tissues is relatively low; and it undergoes a fast metabolism in the body, resulting in a greatly reduced biological activity.

EGCG bioavailability follows the Lipinski’s rule of five, i.e., its bioavailability is determined by molecular size, apparent size (formation of hydrated shells) and polarity [25]. For example, studies have shown that EGCG (relative molecular mass, 458.38 g/mol; eight phenolic hydroxyl groups) has a much higher bioavailability than (-)-Epicatechin (relative molecular mass, 290.27 g/mol; five phenolic hydroxyl groups), and the sizeable hydrated shell formed by hydrogen bonding between water molecules and EGCG reduces its absorption in the body [26]. In general, high oral absorption and bioavailability do not occur with EGCG, and the highest catechin plasma concentration of 1–2 mol/L can be achieved within 1–2 h after ingestion; rapid absorption then occurs within 24 h after the initial administration [27], and is accompanied by evident reversion of plasma concentration to baseline. A variety of factors inhibit the bioavailability of EGCG including internal factors such as EGCG sensitivity to digestive tract conditions, poor intestinal transport, and rapid metabolism and clearance; and external factors such as effects of intake, food matrix, nutritional status, etc. [28,29]. Synergistic or restrictive effects also occur among various factors.

EGCG can perform a variety of secondary metabolic reactions in the body, such as methylation, sulfation, and glucuronidation. Oral EGCG mainly enters the stomach and small intestine through the esophagus and is absorbed into blood via the small intestinal mucosa. It is then converted into various metabolites via enzymatic reactions in the liver and exerts different health and pharmacological effects from the heart to various tissues and organs throughout the body. Its final metabolite is then excreted through urine or fecal discharge [30]. In brief, limitations such as low bioavailability, fast metabolism and short half-life pose significant challenges in the assessment of EGCG activity in research and drug development.

### 2.3. Safety of EGCG

EGCG possesses antioxidant and pro-oxidative properties [31]. It oxidizes in cell culture media and produces reactive oxygen species (ROS) that lead to cell death [32,33]. At low doses (EGCG < 67.8 mg/kg) [34], EGCG does not induce ROS production and organ toxicity, and the intensity of its activity is related to the treatment used. Achieving a good result often requires high doses; however, high doses of EGCG triggers a series of toxic side effects [35]. First and second phase clinical studies found that symptoms of EGCG use include: hepatotoxicity, nausea, insomnia, abdominal pain and diarrhea etc. [36,37,38], of which hepatotoxicity is a toxic side effect reported during the use of high dose EGCG [39]. ROS can also activate redox-sensitive transcription factor nuclear factor erythrocyte 2-related factor 2 (Nrf2), thereby upregulating the antioxidant response element pathway [40]. It has been reported that lower levels of ROS produced by moderate doses of tea polyphenols activate Nrf2 to inhibit oxidative stress, but high doses of EGCG (>750 mg/kg) produce high levels of ROS to induce toxicity [41]. In conclusion, EGCG has acceptable safety. Low to medium doses of EGCG have little harm to the human body. It can prevent the occurrence of digestive tract tumors and can be used as dietary additives. High EGCG doses, however, will exhibit many toxic side effects in the body, limiting its development and use. Solving the high dose toxicity of EGCG is a significant issue that must be resolved in future development of EGCG-containing drugs.

## 3. Research Status of EGCG in Digestive Tract Cancer Treatment

Numerous in vitro, in vivo and clinical studies have shown that EGCG can prevent multiple gastrointestinal tumors by binding to the 67 kDa laminin receptor (67LR) on the surface of tumor cells [42]. By inducing oncogene silencing to prevent digestive tract tumors [43], EGCG can regulate cell cycle hormones, cell cycle-dependent protein kinases (CDKs), CDK inhibitors etc. [44], to stop the growth of tumor cells. EGCG can also change survivin expression, phosphatidylinositol-3-kinase (PI3K) and other signaling pathways [45], induce tumor cell apoptosis, regulate matrix metalloproteinase (MMP) activity, interfere with tumor angiogenesis [46] and ultimately inhibit tumor cell invasion and transformation. Below, we briefly discuss the roles of EGCG.

### 3.1. Induction of Tumor Cell Growth

Tumor cells, like normal cells, have a complete growth cycle that is directly regulated by proteins such as Cyclin, Rb and CDKs. EGCG blocks insulin-like growth factors (IGF) binding to its receptor, inhibits the PI3K-Akt signaling pathway, downregulates the phosphorylation of the PI3K-Akt signaling pathway proteins, Akt and mTOR, and inhibits the expression of the downstream cycle-related proteins, Cyclin D1 and CDK [47,48], leading to cell cycle arrest; these activities contribute to tumor cell growth inhibition. EGCG can also induce the conversion of hypophosphorylated cyclin Rb to its hyperphosphorylated form, and upregulate p53 protein expression, leading to tumor cell growth arrest in the G2/M phase and S phase [47] (Figure 2).

EGCG can change the methylation status of p38, mitogen-activated protein kinase (MAPK) and C-Jun N-terminal kinase in the oral tumor cell, CAL-27 [49], and arrest tumor cell cycle growth. In human colorectal cancer HT-29 cells, EGCG inhibits the expression of Cyclin D1 protein; stops the cell cycle in the G1 phase [50]; downregulates the Bcl-2 protein; upregulates Bax, caspase-3, and caspase-9 [51]; induces cytochrome c release; damages the mitochondrial membrane; and promotes apoptosis of cancer cells. In rat experiments, EGCG significantly inhibited nitromethylbenzylamine (NMBA)-induced esophageal cancer [52], and promoted prostaglandin E2 (PGE2) by upregulating the expression level of the rate-limiting enzyme, cyclooxygenase-2 (COX-2), and PGE2 synthesis, arresting the growth cycle of esophageal tumor cells.

### 3.2. Induction of Tumor Cell Apoptosis

Apoptosis is an orderly protective mechanism of cells where abnormal or damaged cells are removed from the body. Apoptosis plays a vital role in cell development and metabolism. There are two main pathways to apoptosis: the death receptor pathway (external path) and the mitochondrial pathway (intrinsic pathway); however, inducing tumor cell apoptosis is the primary route in tumor removal [53].

EGCG can induce apoptosis of gastrointestinal cancer cells in a dose-dependent manner, without affecting the growth of healthy cells [54]. It can also result in apoptosis of cancer cells via various pathways (Figure 2): (1) EGCG can bind to the anti-apoptotic proteins, Bcl-2 and Bcl-XL, to induce apoptosis [55]. (2) It can inhibit the activity of AP-1 and I-kappa B, upregulate the transcriptional activity of *TP53*, promote p53 expression, and ultimately, increase the Bax/Bcl-2 ratio to promote cell apoptosis [20]. (3) EGCG can inhibit the signal pathways mediated by TNF-α and LPS, block the activation of NF-κB, activate caspase, and induce enzymatic hydrolysis of the NF-κB/P65 subunit, which destroys its structural domain, resulting in the loss of its trans-activation function, thereby inducing cell apoptosis [56,57]. (4) EGCG increases the amount of cytochrome Cyt-c entering the cytoplasm of the inner mitochondrial membrane of tumor cells, inhibits ATP formation, destroys the membrane potential of the mitochondrial membrane, activates caspase, and promotes the apoptosis of tumor cells [58]. (5) EGCG can also bind to the 67 kDa receptor on the surface of tumor cells, and inhibit the phosphorylation of extracellular signal-regulated protein kinase (ERK)1/2, p38, and JNK mediated by peptide glycan (PGN), inducing apoptosis [59].

In the colorectal cancer cell, HT-29, EGCG inhibits LPS-mediated IKK phosphorylation, blocks the expression of NF-κB protein [60,61], induces the FLAs-mediated JNK signaling pathway, and destroys mitochondrial membrane potential, changing its permeability for the release of Cyt-c [62,63]; this activates the activities of caspase-3 and caspase-9, leading to DNA activation in colorectal cancer cells. Fragmentation and nucleus coagulation can promote cell apoptosis. After treating the esophageal cancer cell line, ECa109, with EGCG, a decrease in cell viability [64], increase in the apoptotic rate, and demethylation, downregulation and protein expression of the *p16* gene [65], resulted in the induction of increased ECa109 cell apoptosis. EGCG may induce the downregulation of Bcl-2 family protein expression and the upregulation of Bad and Bax protein expression in gastric cancer MKN45 cells [66,67], increase the Bax/Bcl-2 ratio, lead to the activation of mitochondrial-dependent caspase and apoptotic cell death, and promote the apoptosis of cancer cells. EGCG can also inhibit the expression of oncogenes Id1 and p68 in the gastric cancer cells, AGS and A2521 [14,15], change the related metabolic pathways, and induce apoptosis of cancer cells.

EGCG primarily antagonizes digestive tract tumors by inducing tumor apoptosis. However, the methods of inducing this apoptosis are complex and diverse, and many apoptotic mechanisms remain unclear. One of the leading research directions in future cancer research is the study of the mechanisms of tumor apoptosis to achieve a more effective induction of this process.

### 3.3. Epigenetic Effects

In tumor cells, hypermethylation of the promoter for the tumor suppressor gene, *p16*, is inactivated. This results in either low or no expression of the tumor suppressor gene, which leads to tumor production. In recent years, several studies have shown that EGCG can inhibit the biological activity of DNA methyltransferase and miRNA expression [68], which are valuable tools in tumor protection and treatment. EGCG inhibits the activity of methyltransferases (DNMTs) by scavenging free radicals and displaying antioxidant activities. Demethylation of the tumor suppressor genes, p16 and 21, is caused by hypermethylation in digestive tract tumor cells. When tumor suppressor gene expression is upregulated, tumor cell apoptosis increases and tumor production is inhibited.

EGCG reverses the hypermethylation status of RECK in oral tumor cells, and significantly increases the transcription level of RECK [69]. In human esophageal tumor cells, EGCG can reverse the hypermethylation status of p16, RAR, MGMT and other tumor suppressor genes [70]. Treating the esophageal cancer cell line ECa109 with EGCG resulted in demethylation of the p16 gene and its mRNA. Its protein expression is also significantly increased [64], thereby effectively promoting apoptosis of ECa109 cells and inhibiting tumor cell growth. EGCG can activate some methylation-silencing genes in the human esophageal cancer cell, KYSE510, allowing tumor cells to re-express RARα and HMLH1 proteins to inhibit tumor cell proliferation [71].

EGCG exhibits different genetic effects in normal and colorectal tumor cells. When used to treat the normal human colon epithelial cells, NCM460, and colon adenoma 205 cells (COLO205), chromosome stability of the cells can significantly change, apoptosis can be induced, and cell division can be inhibited [54]. In human colorectal cancer cells such as HCT116 [72], 150 μM EGCG can effectively restore the level of RXRalpha activity in human cells, reduce the methylation of RXRalpha promoter, thereby regulating the reversal and involvement of gene silencing in colorectal cancer. In two models of digestive tract tumors, 0.16% EGCG in drinking liquid had 47% inhibitory effect on small intestinal tumors in Apcmin/+mice, 70% inhibitory effect on esophageal cancer in rats [73], and EGCG effectively inhibited the occurrence of intestinal tumors in mice.

### 3.4. Inhibition of Telomerase Activity

In normal cells, telomeres promote DNA senescence; however, telomerase allows tumor cells to divide indefinitely. Telomerase inhibition can be a potential cancer treatment as it is of great significance in the treatment process. EGCG auto-oxidatively decomposes the galloyl group, regulates epigenetics [74], inhibits telomerase activity [75], reduces cell viability, and induces cell senescence and apoptosis. In human SCLC cells treated with 70 μM EGCG for 24 h [76], telomerase activity decreased by 50–60%, and apoptosis became apparent at 36 h of treatment. It is worth noting that although telomerase inhibition is a promising new adjuvant therapy, telomerase activation may be beneficial in some cells.

### 3.5. Inhibition of Tumor Angiogenesis and Metastasis

Tumor blood vessels not only provide oxygen and nutrients for tumor growth and development but creates a pathway for tumor cell metastasis. Tumor angiogenesis is regulated by critical enzymes such as MMP-9. EGCG can inhibit the activation of metalloproteinase-1 (MT1-MMP) by inhibiting MMP-9 [77], to cause subsequent inhibition of angiogenesis and tumor growth. Meanwhile, EGCG can significantly reduce VEGF, bFGF, HIF-1α, and HIF-1β [78]. The expression of tumor cells interferes with the formation of tumor blood vessels [79], and a significant effect is observed following tumor treatment. EGCG can also block VEGF signal transduction pathways [80], such as ERK phosphorylation, which induces the phosphorylation-deficient mutants of FOXO to produce FOXO transcriptase activity and inhibit HUVEC cell migration and capillary formation [58] (Figure 3).

In mouse colorectal cancer SW837 cells, EGCG can downregulate the expression of HIF-1α, HIF-1β and VEGF in tumor cells [81], reduce tumor vascular density, effectively control tumor cell metastasis and inhibit the PI3K/Akt signaling pathway [82]. In addition, EGCG delivery inhibits the expression of the rapamycin receptor protein (mTOR) and inducible nitric oxide synthase (iNOS), ultimately inhibiting tumor angiogenesis. EGCG can inhibit VEGF/vascular endothelial growth factor receptor (VEGFR) activity in human colorectal cancer cells and reduce the expression of hypoxia-inducible factor (HIF-1α) protein and VEGF that promote angiogenesis [83]. EGCG can also inhibit colon cancer by blocking the activation of the epidermal growth factor receptor (EGFR), insulin-like growth factor-1 receptor (IGF-1R) and VEGFR2 of the RTKs family. Various cancer cells such as liver cancer proliferate and induce apoptosis of cancer cells [84].

In APCM/+ mice intestinal tumors, EGCG induced the ubiquitination of the growth factor, bFGF [85], and reduced its expression to effectively inhibit the growth and metastasis of tumors. Besides, EGCG can significantly inhibit the activity of the transcription activating factor, Stat3, in gastric cancer AGS cells, affect the protein expression of VEGF and its gene, and substantially inhibit the proliferation and metastasis of tumor cells induced by VEGF [86]. EGCG could significantly inhibit the growth of blood vessels in xenograft gastric cancer SGC-79 nude mice, and the microvessel density was also observed to decrease by 38.2% in these mice compared to that in control mice [87]. EGCG was demonstrated to elicit a good effect in clinical and scientific research of hepatocellular carcinoma. In one experiment, HuH7 cells were treated with 100 ug/mL EGCG for 6 h. The results showed that the expression levels of VEGFR-2 and p-VEGFR-2 were inhibited in a dose-dependent manner, thus inhibiting the growth of tumor vessels [88]. At the same time, the nude mice were fed with 0.1% and 0.01% EGCG for 5 weeks [88]. EGCG significantly inhibits tumor growth in nude mice by inhibiting angiogenesis. However, there was no significant difference between 0.1% and 0.01% EGCG inhibition. Altogether, EGCG can effectively mitigate tumor angiogenesis and metastasis, and effectively reduce tumor density.

### 3.6. Other Influences

Studies have shown that inflammation is greatly related to the occurrence, growth and metastasis of tumors, especially because of the swelling observed in digestive tract tumors. Reducing the appearance of redness in the gastrointestinal tract system can effectively prevent the occurrence of digestive tract tumors.

The high concentration of EGCG (0.64%) can effectively remove specific inflammatory mediators, inhibit chemotaxis of chemokine receptor 3 (CXCR3) and its ligands, and play an anti-inflammatory role in preventing tumorigenesis. EGCG can inhibit the expression of the transcription factors NF-κB and COX-2 [89], producing a significant anti-inflammatory effect to effectively inhibit the occurrence of inflammation and reduce tumor incidences. This proves that the excellent anti-inflammatory activity of EGCG may be one of the essential mechanisms in tumor prevention. EGCG treatment to colorectal tumor cells downregulates PI3K/Akt/NF-κB phosphorylation, and p65 acetylation reduces the levels of proinflammatory mediators [90]. Activating ERK1/2 signaling and Nrf2 acetylation increase heme oxygenase-1 (HO-1) expression [91], thereby reducing DSS induced colitis for effective reduction of the incidence of colorectal cancer. EGCG can also effectively improve the occurrence of colorectitis, but its mechanism needs further research.

Metabolism produces a large number of free radicals that damage cells and induce cell carcinogenesis. Cancer cells have a higher level of oxidative stress than normal cells. Therefore, tumor cells may be more sensitive to drugs that abundantly produce ROS or those that impair cell ROS clearance, which result in the death of these cells. EGCG is an excellent antioxidant [92], which can play an antioxidant role by scavenging free radicals and chelating metal ions. Studies have shown that the EGCG-mediated endogenous antioxidant system plays a role in tumor cell apoptosis of colon cancer in xenograft nude mice treated with dietary EGCG [93]. The synergistic action of EGCG with doxorubicin in a chemoresistant model of hepatocellular carcinoma (HCC) resulted in effective tumor cell growth inhibition [94]. The significant antioxidant activity and oxidative damage of EGCG may be the key to its antagonism of tumorigenesis.

## 4. EGCG Interrupts the Tumor Signaling Pathway

Signal transduction is the process whereby extracellular factors bind to cell receptors, triggering a series of biochemical reactions, and even the transcription and expression of cellular genes. EGCG can induce gene regulation disorders in tumor cells, which lead to an abnormal signal transduction network, and tumor cell growth arrest or apoptosis [95,96]. EGCG exerts significant effects on various signal pathways in digestive tract tumors and has the potential for use as a targeted drug for these tumors.

### 4.1. Effects of Epidermal Growth Factor Receptor (EGFR) on Epidermal Growth Factor

EGFR, a cell membrane receptor, is a member of the receptor family, ErbB-1. EGFR is activated after binding to its ligands, EGF and TGF-alpha. EGFR catalyzes enzymes related to inositol phospholipid metabolism, regulates protein phosphorylation which produces a second messenger, and transmits signals to the nucleus along multiple signaling pathways. These activities result in the expression of proto-oncogenes, C-myc and C-fos, in the nucleus, and induces cell proliferation. EGCG can inhibit the binding of EGFR to its receptor by inducing EGFR autophosphorylation [97], and inhibit EGFR internalization, EGFR signal transduction, and tumor cell proliferation and angiogenesis [98].

EGCG inhibited the binding of EGF and EGFR in the esophageal cancer cell, KYSE150, which changed the membrane dimerization and activation of the EGFR tissue [99], inhibited the activation of RTKs related to other membranes, blocked the metabolic pathway to induce cell growth cycle arrest, and ultimately, blocked tumor angiogenesis and metastasis.

### 4.2. Effect on the MAPKs Pathway

The MAPK pathway is the most critical kinase chain in cells, and is mainly comprised of three kinases: protein kinase (ERKs), c-Jun N-terminal protein kinase (JNKs/STAT) and p38 protein kinase; the activation of ERK is related to cell proliferation, JNK is relevant to cell stress and apoptosis, and p38 is linked to inflammatory response.

EGCG can significantly inhibit HGF-induced Met phosphorylation, cell growth and invasion, and the expression of MMP-2 and MMP-9 in oral cancer cells [100]. EGCG blocks HGF-induced c-Met phosphorylation (i.e., phosphorylation of downstream kinases, Akt and ERK), but inhibits p- Akt and p-ERK. EGCG is involved in the increase in p38, JNK, caspase-3 and poly ADP-ribose synthase in the MAPK signaling pathway, and when used to treat colorectal tumor cells, it interferes with MAPK signaling [101], and inhibits COX-2 expression and PGE2 secretion. EGCG inhibits VEGF and the glucose transporter to inhibit tumor cell proliferation and reduce tumor angiogenesis. It can also act on the colorectal cancer cell line, HT-29 [102], causing mitochondrial cell damage and activating JNK-mediated apoptotic cell death. In the human gastric tumor cell, AGS [103], EGCG was shown to inhibit the activation of ERK, JNK and p38 induced by phorbol ester myristate (PMA) in a time-dependent manner, and inhibit downstream nuclear factors, thereby inhibiting digestion and arresting tumor growth.

EGCG inhibits extracellular signal-regulated kinases 1 and 2 (ERK1/2) phosphorylation and p38 MAPK activity. In addition, it has been found to activate three MAPKs (ERK, JNK and p38) in human liver cancer HepG2-C8 cells in dose- and time-dependent manners [104]. Low concentrations of EGCG induce the expression of protein kinase genes through MAPK activation. Although high concentrations also result in MAPK activation, they lead to apoptosis. Therefore, the influence of EGCG on MAPK signal is associated with many controversies, warranting further research.

### 4.3. Effect on the Nuclear NF-κB Pathway

The NF-κB signaling pathway involves kinase-linked activation. Under external stimulation, TRAF2 and RIP (receptor-interacting protein) activate NIK (NF-κB-induced kinase), which then activates IKKα. Activated IKKα phosphorylates IκB-α causing ubiquitination and dependence on the activation of the 26s proteasome activation. NF-κB is isolated and translocated into the nucleus, initiating the transcription of downstream anti-apoptotic genes and pro-inflammatory genes to promote tumorigenesis and metastasis.

EGCG induces the reduction of NF-кB expression and inhibits TNF-α and LPS-mediated activation of the NF-кB signaling pathway in tumor cells. In EGCG-treated cells, the NF-кB inhibitory protein, IкBα, undergoes phosphorylation-triggered degradation. EGCG also inhibits NF-кB pathway-induced apoptosis and is involved in the caspase pathway, which regulates the activity of the caspase protein, inducing NF-кB/P65 subunit digestion to destroy its domain, and promoting loss of the transactivation function to induce apoptosis. In the colorectal tumor cell line, HT-29, EGCG inhibits LPS-mediated IKK phosphorylation, and does not depend on the degradation of NF-κB transcription and subsequent dissociation from the NF-κB protein [105]. EGCG induces tumor cell apoptosis, which may be related to the stability of p53, and downregulates NF-κB activity, leading to the downregulated expression of the anti-apoptotic protein, Bcl-2. EGCG was also found to inhibit the activity of COX-2 and reduce the appearance of PGE2, thereby reducing the epithelial-mesenchymal transition process caused by excessive activation of the NF-κB signaling pathway to inhibit tumor cell invasion [106]. EGCG inhibits NF-κB activity and cytokine-induced Il-8 production and secretion in AGS gastric tumor cells to inhibit gastric cancer cell production [107].

### 4.4. Effects on Other Signaling Pathways

In addition to the above pathways, EGCG can inhibit different signaling pathways such as PI3K-Akt, p53, Nrf2, IGF-1 and MMPS protease activity in vitro, to induce growth arrest of gastrointestinal cancer cells for apoptosis, and inhibit tumor angiogenesis and metastasis.

EGCG was demonstrated to downregulate protein serine/threonine phosphorylase 2A (PP-2A) in a dose-dependent manner, affecting its dephosphorylation at the serine 15 site of p53. Such action led to a high expression of the p53 protein, upregulated expression of the apoptotic gene, Bak, and the promotion of cell apoptosis [108]. EGCG has also been shown to upregulate p21 in cancer cells, induce cell cycle arrest and inhibit cancer cell growth. In colorectal cancer, Nrf2-mediated heme oxygenase-1 (HO-1) overexpression result in cell proliferation and apoptosis resistance, and administering a high dose inhibits Nrf2 nuclear localization, cells expressing high amounts of Nrf2 and Nrf2, Nrf2 entry into the nucleus, Nrf2 binding to ARE, and HO-1-ARE promoter activity, inducing cell apoptosis [109]. EGCG can also regulate FOXO transcription factors to repair damaged DNA and control apoptotic genes to induce apoptosis [110].

## 5. EGCG Derivatives Antagonize Gastrointestinal Tumors

Enhancing the stability and bioavailability of EGCG has become a hotspot in research and pharmacological applications of EGCG. In recent years, many researchers have improved the absorption rate and degradation rate of EGCG by nano-technology and liposome technology [29], which in turn, have enhanced its biological activity. Experiments show that the use of nano-scale chitosan to encapsulate catechins can significantly improve the pharmacokinetic characteristics of EGCG and its bioavailability by increasing its absorption rate in the small intestine, thereby greatly enhancing its potency. EGCG NPs can significantly change the pharmacokinetic characteristics of EGCG and increase its bioavailability by more than 2.4 times [29,111,112]. The nanoliposome modification of EGCG significantly enhanced the stability of EGCG, effectively slowed down the changes of antioxidant activity of EGCG in vitro and effectively enhanced the sustained release ability of EGCG [113,114].

Matsumura [115] used molecular modification to prepare the derivatives, EGCG-c4 (butylphthalide), EGCG-c8 (butylphthalide) and EGCG-16 (palm-shelved phthalide), with EGCG as the precursor. When examined, the stability of these derivatives was enhanced. After administering EGCG to mice with the tumor, an increase in the growth inhibition of colorectal cancer cells was observed, with EGCG-16 demonstrating the most apparent inhibitory effect. Superacetylation Protects EGCG (Pro-EGCG) can strengthen the stability of EGCG, enhance its bioavailability, inhibit proteasome activity and induce apoptosis in KYSE150 human esophageal cancer and HCT116 human colorectal cancer cells; this is achieved by Bax upregulation and apoptosis induction [116]. Embedding EGCG in chitosan nanoparticles (NPs) can also improve the stability of EGCG in vivo, significantly increase its intestinal absorption rate and enhance its bioavailability [112]. A study found that nano-EGCG can exhibit a more effective induction of cancer cell apoptosis by increasing ROS/RNS levels [117], decreasing mitochondrial membrane potential and enhancing the therapeutic effect of EGCG on tumors.

Structural changes in EGCG have shown promising results in its antibody action. Other analogs of EGCG, however, require optimization and evaluation to develop more effective, stable and specific analogs as potential new anticancer agents. A major issue in tumor treatment is unraveling how to integrate new molecular results into clinical practice, and more extensive preclinical studies are required to prove the efficacy of these drugs in digestive tract tumors. EGCG analogs have been demonstrated as very important drugs in the prevention and treatment of gastrointestinal cancer. Nano-formulations with increased EGCG bioavailability, and targeted therapy of gastrointestinal tumors require further research to enhance EGCG stability and bioavailability, which will lay the foundation for the development of EGCG as an anticancer drug.

## 6. EGCG Combined with Other Drugs to Antagonize Gastrointestinal Cancer

EGCG can be combined with 5-fluorouracil (5-FU) [118,119], ginseng diol [120], curcumin [121], and other drugs to enhance its efficacy when used to treat digestive tract tumor cells [18,122]. Such combinations may also reduce the adverse reactions of cellular chemotherapy drugs and resistance of tumor cells to chemotherapy drugs. In Table 1, we have briefly summarized the role of EGCG when synergistically applied with other medications to antagonize digestive tract tumors.

EGCG can inhibit EGFR and MMP-2 phosphorylation, decrease MMP-2 activity and level, and enhance the anti-metastasis effect of gefitinib in the human oral squamous cell carcinoma cell, CAL-27. Combining EGCG and natural sodium can significantly improve the pharmacokinetics of the two drugs in the human body [123], and in addition to inhibiting the growth of colorectal cancer in the SDCSC xenotransplantation model, the combination of EGCG and 5-FU was used as an adjuvant therapy with conventional chemotherapy drugs for CRC patients to reduce the sensitivity of tumor cells to chemotherapy drugs. Combination of the vitexin-2-O-xyloside, 4-methylthioalkyl-3-butenyl isothiocyanate, and EGCG induces the accumulation of many ROS in colorectal cancer cells and promotes mitochondrial pathway activation to induce apoptosis [124]. The combination of EGCG and sulindac showed a synergistic effect in the prevention of colorectal cancer caused by oxidized methane (AOM) [125]. Reduced incidence of colorectal cancer, an increase in tumor cell apoptosis, and reduced adverse reactions occur by sulindac-only treatment. EGCG administration to genistein-treated APC (min/+) mice showed that genistein increased the levels of EGCG in the small intestine and plasma [126], and significantly enhanced the cytoplasmic level and growth inhibitory activity of EGCG on human colorectal cancer cells.

It is also worth noting that some studies have shown that EGCG may also interact with other anticancer drugs, and this action can reduce either the drug’s anti-tumor activity and bioavailability. For example, the combination of gefitinib and EGCG reduces the bioavailability of gefitinib [127]. Further research is therefore needed to determine the interaction of EGCG with other drugs when used to antagonize digestive tract tumors, and to provide a safe and effective combination strategy for clinical use.

## 7. Conclusions

The antagonizing mechanism of EGCG in gastrointestinal cancer involves inhibiting tumor stem cell proliferation, reducing the incidence of inflammation, preventing tumor production, regulating cell signal transduction, inducing cell growth cycle arrest and apoptosis, inhibiting tumor angiogenesis, and blocking metastasis and invasion of tumor cells. When EGCG is combined with other anticancer drugs, it can play a synergistic anticancer role, reverse drug resistance of tumor cells, and reduce the probability of tumor recurrence after surgery. Based on these actions, EGCG immensely demonstrates the potential to antagonize digestive tract tumors and has broad prospects for development. The mechanism of EGCG antagonizing gastrointestinal cancer is not yet perfect. These mechanisms include interactions with many cellular signaling pathways that are important for cancer cell function. This interaction appears to alter these signaling pathways and inhibit the effects of cancer cells. However, how EGCG interacts with cancer cells, its cellular targets and the exact mechanism of action remain to be established. This important information helps to elucidate the mechanism of action of EGCG as a novel chemopreventive agent. In addition, the antagonistic effects and mechanisms of EGCG derivatives, such as theaflavins and thearubins on digestive tract tumors need to be further studied. The optimal combination of EGCG and mixed polyphenols against digestive tract tumors still needs to be verified.

The use of EGCG as an anticancer drug in clinical applications has many challenges. As a chemopreventive agent, EGCG must be ingested over a long period and exceed the average dietary intake, without causing side effects. Since the potential toxicity associated with this regimen has not been studied, it is unknown whether the effective dose of chemoprophylaxis differs from the overexpressed toxicity dose. The use, purpose and method of high non-physiological concentrations of EGCG in a cell culture raise concerns regarding its actual relevance, practical significance and functional relevance, respectively, in in vivo studies. The reduced stability of EGCG and its low oral bioavailability limit its role in preventing the inhibition of digestive tract tumors. These limitations indicate that there is still a long path ahead before EGCG is used in a clinical setting as a therapeutic agent for digestive tract tumors. Future research should focus on large-scale animal experiments and randomized, double-blind clinical studies to further examine and verify the antagonistic mechanism of EGCG on gastrointestinal tumors, develop EGCG derivatives, improve EGCG stability and improve its oral bioavailability. Performing such tasks will provide sufficient theoretical basis for chemotherapy of digestive tract tumors and lay the foundation for future developments of a new generation of EGCG anti-tumor drugs.

## Figures and Tables

**Figure 1 molecules-24-01726-f001:**
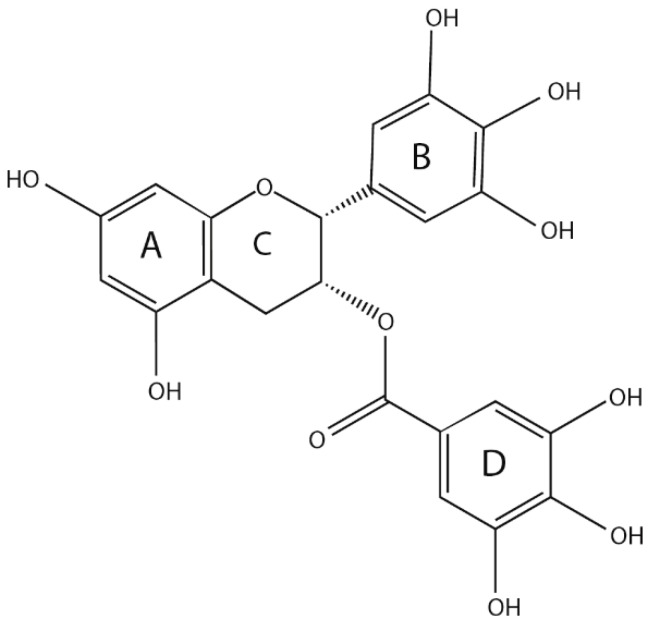
Structure of epigallocatechin-3-gallate (EGCG).

**Figure 2 molecules-24-01726-f002:**
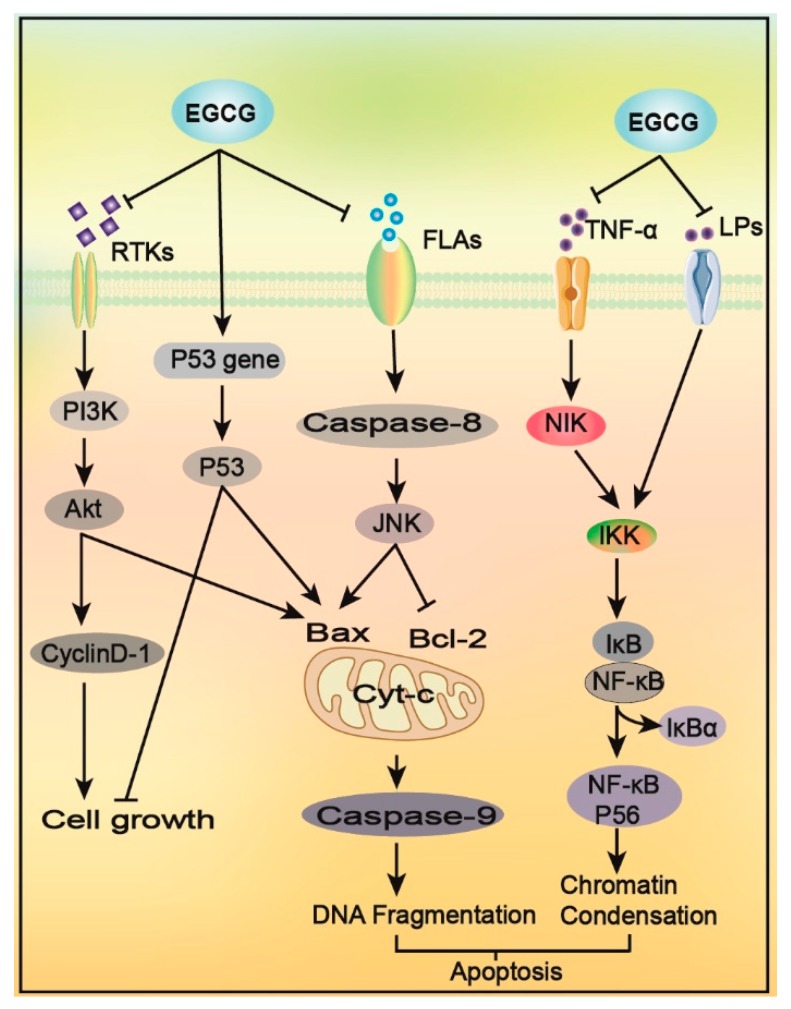
The mechanism used by EGCG to promote cancer cell growth arrest and apoptosis. EGCG blocks the the phosphatidylinositol 3 kinase (PI3K)-Akt pathway and downregulates Cyclin D-1 expression, leading to cell cycle arrest. This effect can also be achieved by the upregulation of p53 expression. EGCG inhibits FLAs-mediated JNK signaling pathway, increases the ratio of Bax/Bcl-2, and promotes apoptosis. Other roles of EGCG include promoting the increase of Cyt-c in the mitochondrial inner membrane, destroying the membrane potential of the mitochondrial membrane, activating caspase, and promoting tumor cell migration and apoptosis. EGCG also inhibits TNF-α, LPS and other mediated inflammatory signaling pathways, blocks NF-κB activation, induces NF-κB/P65 subunit digestion, disrupts its domain, promotes cancer cell apoptosis and reduces validation response caused by somatic apoptosis.

**Figure 3 molecules-24-01726-f003:**
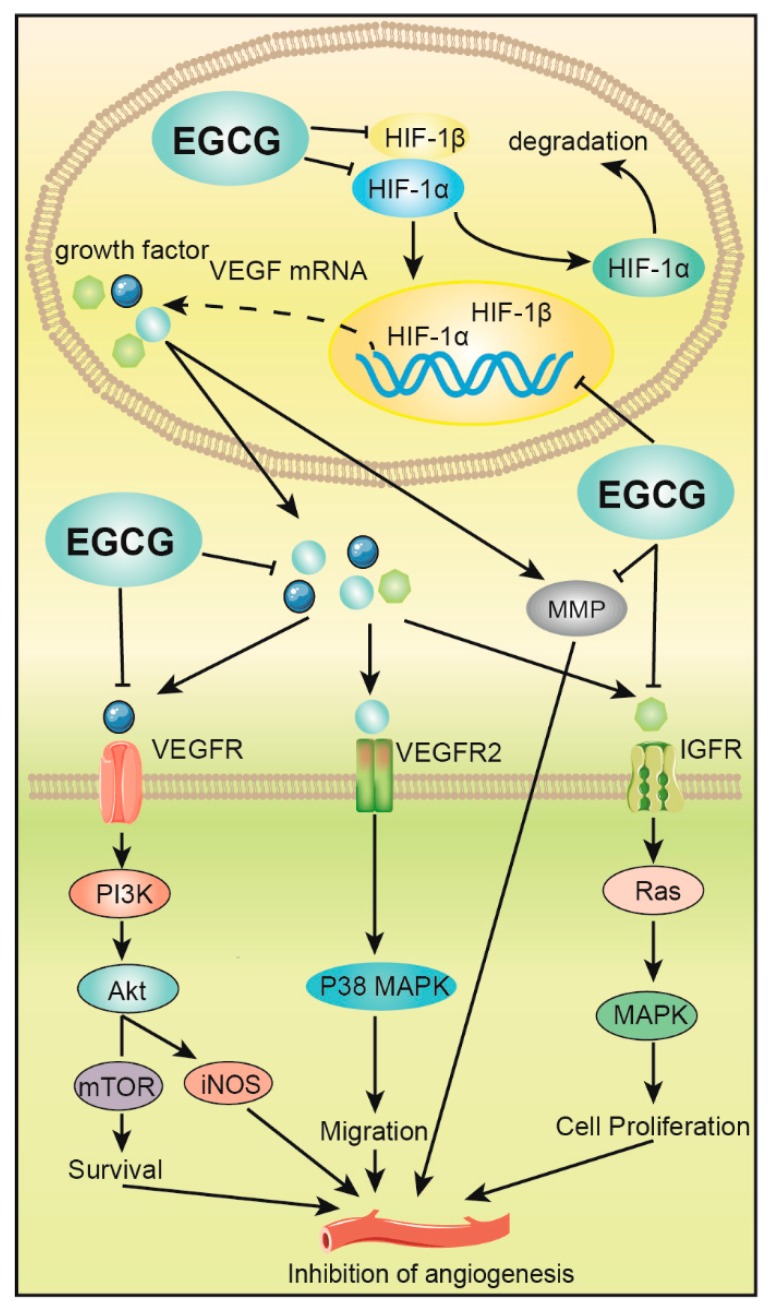
EGCG inhibits angiogenesis of digestive tract tumors. EGCG inhibits the expression of hypoxia-inducible factor (HIF-1α and HIF-1β), which downregulates VEGF expression, and ultimately inhibits the formation and metastasis of tumor vasculature by mediating PI3K/Akt, p38 mitogen-activated protein kinase (MAPK) and MAPK signaling pathways. EGCG also inhibits the activation of enzymes such as matrix metalloproteinase (MMP), thereby inhibiting angiogenesis and growth.

**Table 1 molecules-24-01726-t001:** Synergistic anticancer effects of EGCG and other drugs.

Drugs	Model	Synergy Effect	Concentration Ranges	Mechanism	Reference
5-FU	Gastric cancer HCC cells	↓Growth stagnation, ↑Apoptosis, ↓Anti-sensitivity, augmented the anti-tumor effect of 5-FU in Hep3B cells	EGCG 5 μmol/L, 20 μM	↓REK, ↓Akt, Bcl-2, ↑Bax, MDR-1 expression,	[118,119]
Panax	Colon HCT-116, SW-480	↑Apoptosis,	20 μm Panaxadiol and 20 and 30 µm of EGCG	↑ROS, Activating NF kappa B pathway	[120]
Curcumin, lovastatin	Xenotransplantation model for esophageal carcinoma	↓esophageal cancer cell growth	lovastatin (4 µmol/L), curcumin (40 µmol/L), EGCG (40 µmol/L)	↓Phosphorylation of ERK1/2, c-jun and COX-2 expression	[121]
Gefitinib	Mouth cancer	↓Growth stagnation	gefitinib (10 μM) and EGCG (25–100 μM)	↓Metastasis of tumor cells	[123]
Vite -2-*O*-glycoside (4-methyl trialkyl-3- butenyl isothiocyanate)	LoVo and CaCo-2	↓Growth stagnation, ↑Apoptosis	vitexin-2-*O*-xyloside (40 μg/mL), 4-methylsulphanyl-3-butenyl isothiocyanates (5 μg/mL) and EGCG (10 mg/mL)	↑ROS, ↑Apoptosis	[124]
Sullin acid	Colitis	Prevention	sulindac (20 mg/kg) EGCG(2.5 mL)	↑Apoptosis	[125]
Genistein	HT-29	↓Growth stagnation	EGCG (75 mg/kg) and genistein (200 mg/kg)	↑Bioavailability	[126]
Sunitinib	Human	↓Bioavailability	EGCG solution (8 mg/mL), sunitinib solution (4 mg/mL)	Precipitate	[127]

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
