# Peer review of "Advances in the Antagonism of Epigallocatechin-3-gallate in the Treatment of Digestive Tract Tumors"

_molecules, 2019, doi:10.3390/molecules24091726_

Round 1

Reviewer 1 Report

The authors of the manuscript summarize the literature on epigallocatechin-3-gallate (EGCG), its properties, activity and its use in cancer treatment. The manuscript is clearly written and well arranged in the sections and the sub-sections with the consistent layout, however sometimes is easy to get lost because of the density of the information. In my opinion, more figures and/or tables are needed to improve the reading and the organize the information provided.

It is worth noting that this review was prepared based on the latest literature, the vast majority of referred papers were published during the last 10 years and very few of 115 cited papers were published before 2010 year.

Minor comments

1. Lines 53 and 136. “cancer[8]” and “tumors[36]”please insert a space between them.

2. Conclusion section should be expanded.

Author Response

Response to Reviewer 1:

Point 1: The authors of the manuscript summarize the literature on epigallocatechin-3-gallate (EGCG), its properties, activity and its use in cancer treatment. The manuscript is clearly written and well arranged in the sections and the sub-sections with the consistent layout, however sometimes is easy to get lost because of the density of the information. In my opinion, more figures and/or tables are needed to improve the reading and the organize the information provided.

Response 1: Thanks very much for your kind suggestion, we describe the usage of EGCG in this paper and insert Figure 4 (Structure Formula of EGCG Derivatives.). Hope to improve the readability of the article.

Point 2: It is worth noting that this review was prepared based on the latest literature, the vast majority of referred papers were published during the last 10 years and very few of 115 cited papers were published before 2010 year.

Response 2: Yes, the newest references will provide the current research hotspots, trends and future research directions of EGCG antagonism to digestive tract tumors, so the new references are very important. On the other hand, researchs before 2010 is still valuable and we added some files also, it is hoped that the researchs on the antagonism of EGCG against digestive tract tumors will be reviewed comprehensively.

Point 3: Minor comments

1. Lines 53 and 136. “cancer[8]” and “tumors[36]”please insert a space between them.

2. Conclusion section should be expanded.

Response 3:

1. Errors in Lines 53 and 136 have been corrected, thanks for your notification.

2. The conclusion has been expanded. The revised conclusions are as follows:

The antagonizing mechanism of EGCG in gastrointestinal cancer involves inhibiting tumor stem cell proliferation, reducing the incidence of inflammation, preventing tumor production, regulating cell signal transduction, inducing cell growth cycle arrest and apoptosis, inhibiting tumor angiogenesis, and blocking metastasis and invasion of tumor cells. When EGCG is combined with other anticancer drugs, this can enhance the anticancer effect of both drugs in the combination, reverse drug resistance of tumor cells, and reduce the probability of tumor recurrence after surgery. Based on these actions, EGCG immensely demonstrates the potential to antagonize digestive tract tumors and has broad prospects for development. The mechanism of EGCG antagonizing gastrointestinal cancer is not yet perfect. These mechanisms include interactions with many cellular signaling pathways that are important for cancer cell function. This interaction appears to alter these signaling pathways and inhibit the effects of cancer cells. However, how EGCG interacts with cancer cells, its cellular targets and the exact mechanism of action remain to be established. This important information helps to elucidate the mechanism of action of EGCG as a novel chemopreventive agent. In addition, the antagonistic effects and mechanisms of EGCG derivatives, such as theaflavins and thearubins on digestive tract tumors need to be further studied. The optimal combination of EGCG and mixed polyphenols against digestive tract tumors still needs to be verified.

The use of EGCG as an anticancer drug in clinical applications has many challenges. As a chemopreventive agent, EGCG must be ingested over a long period and exceed the average dietary intake, without causing side effects. Since the potential toxicity associated with this regimen has not been studied, it is unknown whether the effective dose of chemoprophylaxis differs from the overexpressed toxicity dose. The use, purpose and method of high non-physiological concentrations of EGCG in a cell culture raise concerns regarding its actual relevance, practical significance and functional relevance, respectively, in vivo studies. The reduced stability of EGCG and its low oral bioavailability limit its role in preventing the inhibition of digestive tract tumors. These limitations indicate that there is still a long path ahead before EGCG is used in a clinical setting as a therapeutic agent for digestive tract tumors. Future research should focus on large-scale animal experiments and randomized, double-blind clinical studies to further examine and verify the antagonistic mechanism of EGCG on gastrointestinal tumors, develop EGCG derivatives, improve EGCG stability, and improve its oral bioavailability. Performing such tasks will provide sufficient theoretical basis for chemotherapy of digestive tract tumors and lay the foundation for future developments of a new generation of EGCG anti-tumor drugs.

Reviewer 2 Report

Decision on Molecules “Advances in the antagonism of 2 epigallocatechin-3-gallate in the treatment of 3 digestive tract tumors”: Review.

General comments:

The review article focused on epigallocatechin-3-gallate (EGCG) and its advances to treat digestive tract tumors. Covering the potential of ECGC to antagonize and prevent tumors of the digestive tract and its limitations of low bioavailability, poor stability and the need for future clinical trials. However, and most importantly it lacked quantitative values/detail required for a review article to showcase the current status of literature findings and future directions. The studies referenced and described in the review must include a point of comparison and to describe the context of the measurements with a set of quantitative values and details, for example Table 1 has potential to include such details. The lack of comprehensive scientific evidence supported by quantitative details gives an appearance of a subjective review article. At this point due to the lack of comparative quantitative values the decision of “review” was reached.

Specific comments:

·       Line 52 “offend” please replace with interrupt or delay the onset of.

·       Line 55 “contributed to its attraction of research interest” please replace with “attracted research interest”.

·       Lines 62-63 needs a cited reference/s.

·       Lines 65: clarify with more detail to understand/provide the context "safety" is described.

·       Line 66: include concentration range/s of ECGC quantified within tea with cited references and then link the concentration to a cost.

·       Lines 80-81: requires a cited reference.

·       Lines 80-85: requires cited references.

·       Lines 98-102: requires cited references.

·       Line 123: define and clarify with quantitative measurements of “highly safe” also describe as “safe” and state the limits of which safe is defined.

·       Line 124: “High ECGC doses” again add quantitative value/details.

·       Lines 1280-130: If you want to use "numerous" at the beginning of this sentence you will need to add more than one reference here - unless this reference has many references supporting this statement (if so please use "[36] and references therein]").

·       Line 132: Please detail the abbreviation before use.

·       Lines 223-225 - This requires more description, e.g. the dose concentrations, duration of the dose period, length of the trial (days/months?), how many humans participated and the efficacy?

·       Lines 271-274: More detail again - to specify efficacy, which dosage concentration was most effective, how did it compare to the placebo/control group it may be good to include a figure here to represent the main findings of this study.

·       Line 280: please add value to describe “High concentration”.

·       Line 294: please add detail/context for the use of excellent or delete.

·       Lines 303-308: requires references.

·       Lines 384-387: very important to highlight the importance of the review, as one of the limitations is the bioavailability - please add references here and more detail with representative/measured values of concentrations, time periods and efficacy.

·       Lines 412-416: This table lacks a lot of detail. Please improve for example specify the concentration ranges and some value of comparison to illustrate the efficacy of the synergistic effects.

Author Response

Response to Reviewer 2: Point 1: General comments: The review article focused on epigallocatechin-3-gallate (EGCG) and its advances to treat digestive tract tumors. Covering the potential of ECGC to antagonize and prevent tumors of the digestive tract and its limitations of low bioavailability, poor stability and the need for future clinical trials. However, and most importantly it lacked quantitative values/detail required for a review article to showcase the current status of literature findings and future directions. The studies referenced and described in the review must include a point of comparison and to describe the context of the measurements with a set of quantitative values and details, for example Table 1 has potential to include such details. The lack of comprehensive scientific evidence supported by quantitative details gives an appearance of a subjective review article. At this point due to the lack of comparative quantitative values the decision of “review” was reached. Response 1: Thanks for your good suggestion. We have revised some quantitative value/details required to demonstrate the current status and future direction of the research results in the literature. Meanwhile, the concentration range of EGCG in Table 1 was also modified, we hope this will ensure the objectivity of the article. Specific comments: Point 2: Line 52 “offend” please replace with interrupt or delay the onset of. Response 2: Line 52 "Violation" has been replaced by interrupt. Point 3: Line 55 “contributed to its attraction of research interest” please replace with “attracted research interest”. Response 3: Thanks, line 55 “contributed to its attraction of research interest” is replaced with “attracted research interest”. Point 4: Lines 62-63 needs a cited reference/s. Response 4: Thanks, lines 62-63 has been inserted a cited reference. Point 5: Lines 65: clarify with more detail to understand/provide the context "safety" is described. Response 5: Thanks, the safety of EGCG was described. The use of medium and low doses of EGCG did little harm to human body, while liver toxicity is caused by high dosage. At the same time, the concentration range of low dose and high dose is described. Point6: Line 66: include concentration range/s of ECGC quantified within tea with cited references and then link the concentration to a cost. Response 6: Thanks, line 66 has been inserted a cited reference. The concentration of EGCG in green tea has been mentioned below. It can be easily obtained from green tea. Catechin is the main secondary metabolite of Camellia sinensis (L.) O. Kuntze, accounting for an estimated 12% to 24% of the dry weight of tea, and EGCG is the primary content, accounting for approximately 50-80% of the total amount of catechins. Point 7: Lines 80-81: requires a cited reference. Response 7: Thanks, lines 80-81 has been inserted a cited reference. Point8: Lines 80-85: requires cited references. Response 8: Thanks, lines 80-85 has been inserted a cited reference. Point 9: Lines 98-102: requires cited references. Response 9: Thanks, lines 98-102 has been inserted a cited reference. Point 10: Line 123: define and clarify with quantitative measurements of “highly safe” also describe as “safe” and state the limits of which safe is defined. Response 10: EGCG has acceptable safety. Low to medium doses of EGCG ( 750 mg/kg· Point 12: Lines 1280-130: If you want to use "numerous" at the beginning of this sentence you will need to add more than one reference here - unless this reference has many references supporting this statement (if so please use "[36] and references therein]"). Response 12: Thanks, references have been added. Point 13: Line 132: Please detail the abbreviation before use. Response 13: Thanks, explanation of abbreviations. Point 14: Lines 223-225 - This requires more description, e.g. the dose concentrations, duration of the dose period, length of the trial (days/months?), how many humans participated and the efficacy? Response 14: Thanks, the concentration and effect of EGCG were explained. Point 15: Lines 271-274: More detail again - to specify efficacy, which dosage concentration was most effective, how did it compare to the placebo/control group it may be good to include a figure here to represent the main findings of this study. Response 15: Thanks, the concentration and effect of EGCG were explained. Point 16: Line 280: please add value to describe “High concentration” Response 16: Thanks, “High concentration” >750mg/kg. Point 17: Line 294: please add detail/context for the use of excellent or delete. Response 17: Thanks, delete excellence, add details. Point 18: Lines 303-308: requires references. Response 18: Thanks, lines 303-308 has been inserted a cited reference. Point 19: Lines 384-387: very important to highlight the importance of the review, as one of the limitations is the bioavailability - please add references here and more detail with representative/measured values of concentrations, time periods and efficacy. Response 19: Thanks, lines 384-387 has been inserted a cited reference, concentration efficiency was described. Point 20: Lines 412-416: This table lacks a lot of detail. Please improve for example specify the concentration ranges and some value of comparison to illustrate the efficacy of the synergistic effects. Response 20: Thanks and we modified the concentration and synergistic effect of EGCG in Table 1.

Reviewer 3 Report

The review paper “Advances in the antagonism of eppigallovatechin-3-gallate in the treatment of digestive tract tumors” is well written and well-structured in term of addressed points. The authors discussing the pros and cons of the research outcomes related with the eppigallovatechin-3-gallate in the treatment of digestive tract tumors and indicate the state of the art in each item. Is therefore my opinion that this review can be published in the present state.

Author Response

Response to Reviewer 3: The review paper “Advances in the antagonism of eppigallovatechin-3-gallate in the treatment of digestive tract tumors” is well written and well-structured in term of addressed points. The authors discussing the pros and cons of the research outcomes related with the eppigallovatechin-3-gallate in the treatment of digestive tract tumors and indicate the state of the art in each item. Is therefore my opinion that this review can be published in the present state. Response: Thanks very much for your kind comments.

Round 2

Reviewer 2 Report

Accept.